# Ruptured Posterior Cerebral Artery Dissecting Aneurysm After Trauma: A Case Report and Literature Review

**DOI:** 10.3390/life16010034

**Published:** 2025-12-25

**Authors:** Chun-Han Chang, Yuan-Yun Tseng, Tao-Chieh Yang

**Affiliations:** 1School of Medicine, Chung Shan Medical University, Taichung City 402, Taiwan, China; hannah348316@gmail.com; 2Department of Neurosurgery, New Taipei Municipal Tu-Cheng Hospital (Built and Operated by Chang Gung Medical Foundation), New Taipei City 236, Taiwan, China; britsey@tmu.edu.tw; 3College of Medicine, Chang Gung University, Taoyuan City 333, Taiwan, China; 4Department of Neurosurgery, Chung Shan Medical University Hospital, Taichung City 402, Taiwan, China

**Keywords:** intracranial dissection aneurysms, lumbar burst fracture, head trauma, subarachnoid hemorrhage, endovascular treatment

## Abstract

Posterior cerebral artery (PCA) aneurysms are rare, accounting for less than 2% of intracranial aneurysms. Among them, dissecting aneurysms frequently occur in the P2 segment. Traumatic PCA aneurysms are extremely uncommon and usually reported in pediatric or young adults following high-energy injuries. We report the case of a 43-year-old woman who sustained a ruptured left PCA P2 dissecting aneurysm with subarachnoid hemorrhage, accompanied by an L2 unstable burst fracture after a high-speed motor vehicle collision. Initial neuroimaging revealed diffuse basal cistern hemorrhage with more predominance at the left side ambient cistern and a fusiform aneurysm with a superimposed saccular component along its anterior portion of left PCA P2 segment. The patient underwent endovascular treatment with a flow-diverting stent and stent-assisted coiling, achieving complete obliteration, followed by lumbar minimally invasive spinal surgery (MISS). The patient recovered without neurological deficits and remained fully independence at a one-year follow-up. Traumatic PCA dissecting aneurysms pose a diagnostic challenge due to their rarity and potential for delayed clinical manifestation, yet they carry a substantial risk of morbidity and rebleeding if untreated. Early recognition through detailed vascular imaging and timely reconstructive endovascular intervention are essential to preventing secondary hemorrhage and optimizing clinical outcomes. This case underscores the need for heightened suspicion for vascular injury in patients with significant craniovertebral trauma.

## 1. Introduction

Dissecting PCA aneurysms may occur due to trauma or arise spontaneously. Traumatic PCA aneurysms are very rare and are most seen in the pediatric or young adults. They are easy to diagnose late [1,2,3]. The mechanisms of traumatic PCA aneurysms are frequently associated with blunt head trauma and may arise from high-velocity impact, such as vehicular accidents or falls, leading to direct mural injury or acceleration–deceleration against the tentorium-induced vessel wall injuries [2,3,4,5].

The case report describes a middle-aged woman who suffered a ruptured left PCA P2 segment anterior portion dissection aneurysm and an L2 lumbar unstable burst fracture due to a car accident on the highway. This article talks about the clinical presentation, imaging results, treatment, follow-up, and prognosis. It is augmented by an examination of possible etiologies, mechanisms, and therapeutic approaches grounded in pertinent literature.

## 2. Case Report

A 43-year-old woman was brought to our emergency department after losing consciousness in a high-speed highway collision in which her vehicle rear-ended another car. Surveillance footage from her dashboard camera showed no direct impact from a hard object; instead, the force of the collision appeared to be absorbed primarily by her torso. She had no known history of hypertension or cerebrovascular disease and there was no family history of intracranial aneurysms, Ehlers–Danlos Syndrome, or Marfan syndrome.

On arrival, her vital signs in the emergency room were as follows: blood pressure at 145/92 mmHg, heart rate at 95 beats per minute, and respiratory rate at 15 breaths per minute. She had regained consciousness with a Glasgow Coma Scale score of E4V4M6. The patient reported retrograde amnesia, headache, and severe lower back pain. No focal neurological deficits were identified on initial examination.

A non-contrast brain computed tomography (CT) scan revealed diffuse subarachnoid hemorrhage (SAH) with a thickness exceeding 1 mm within the basal cistern, predominantly along the left side ambient cistern (Figure 1a). Subsequent CT angiography (CTA) was performed to investigate the etiology of the SAH and demonstrated an irregular vessel contour in the left PCA P2 segment, with a focal arterial pouch consistent with a dissecting aneurysm with a saccular rupture component (Figure 1b). Evaluation of the lumbar spine using CT (Figure 2a) and axial T2-weighted magnetic resonance imaging (MRI) (Figure 2b) revealed an L2 burst fracture with significant spinal canal compromise, resulting in approximately 65% dural sac stenosis. Digital subtraction catheter angiography (DSA) with 3D rotational angiographic acquisition was performed using the Siemens Artis Q angiography system with PURE technology (Siemens Healthineers, Forchheim, Germany). Image reconstruction and aneurysm morphometric quantitative measurements were performed on the Siemens syngo X-Workplace using the syngo 3D Angio (Dyna3D) module. The DSA identified an approximately 5 mm long fusiform dilation along the anterior portion of the left PCA P2 segment (Figure 3a, D1), accompanied by an inferiorly projecting saccular pouch measuring 2.5 mm in diameter (Figure 3a, D2). These findings were diagnostic of a ruptured dissecting aneurysm with a superimposed saccular aneurysm sac (Figure 3a). All cerebrovascular imaging examinations was performed by neuroradiologist Dr. Wu Ming-Ji. A second neuroradiologist Dr. Shen Chao-Yu independently reviewed the images and confirmed the morphometric measurements.

Based on her clinical presentation and radiologic findings, the SAH resulting from rupture of the left PCA P2 segment dissection aneurysm was categorized as Hunt and Hess Grade II [6] and Fisher Grade 3 [7]. The L2 burst fracture was categorized according to the AO Spine Thoracolumbar Classification System as Type A4,N0 [8]. The patient’s initial functional status corresponded to a Barthel Index score of 45 and a modified Rankin Scale (mRS) score of 3.

After multidisciplinary discussion with the neuroradiologists at the Neurovascular Center of Chun Shan Medical University Hospital regarding the prevention of rebleeding from the left PCA dissecting aneurysm, the patient underwent reconstructive endovascular treatment (rEVT) performed by Dr Wu. This intervention was carried out approximately 30 h after the SAH after obtaining informed consent of off-label usage of flow-diverting stent in traumatic cerebral aneurysm from the patient and her family and involved the use of a flow-diverting stents with coiling with significant self-paid expenses. Given the high risk of rebleeding in dissected arteries, along with the potential of in-stent thrombosis and thromboembolic complications following flow-diverting stent deployment, a relatively conservative antiplatelet regimen was selected. The patient received a single loading dose of clopidogrel (Sanofi, Paris, France) 300 mg, administered 8 h before deployment of the device. A Pipeline Flex embolization device (PED Flex device, Medtronic, Irvine, CA, USA) measuring 2.5 mm × 16 mm was positioned across the left PCA P2 segment to reconstruct the parent artery. Subsequently, targeted coil embolization of the saccular pouch in the middle left PCA P2 segment was performed using a stent-assisted coiling technique. The endovascular treatment resulted in near-complete elimination of contrast opacification within the previously visualized dissecting pouch, indicating successful occlusion of the false lumen of the PCA dissection aneurysm (Figure 3b). No immediate procedure-related complications or newly developed neurologic deficits after transarterial embolization (TAE). During the subsequent two weeks after rEVT, the patient exhibited no post SAH sequelae, such as cerebral vasospasm, hydrocephalus, or hyponatremia and maintained antiplatelet therapy with clopidogrel 100 mg daily. The lumbar spinal surgery was performed for L2 burst fracture using a MISS approach without discontinuing antiplatelet therapy. This procedure included left L2 hemilaminectomy for decompression, and percutaneous transpedicle screw fixation from L1 to L3, with preservation of the supraspinous ligament and paraspinal muscle structures.

The patient was maintained on clopidogrel 75 mg daily for one year, with scheduled follow-up imaging that included brain MRI with MRA at three months and one year, as well as a DSA after TAE at six months. These studies were obtained to evaluate interval changes in the configuration of the left PCA aneurysm and to monitor for potential thromboembolic events or in-stent thrombosis following the endovascular procedure. Follow-up brain MRA demonstrated a patent left PCA without evidence of thromboembolism (Figure 4a). The six-month DSA confirmed complete and durable occlusion of the dissection aneurysm, without recurrence or parent artery stenosis (Figure 4b). The patient was discharged approximately one month after the SAH and subsequently recovered well following both the TAE and MISS. At the one-year follow-up, she had a Barthel Index score of 100, and a mRS score of 0, indicating full function independence.

## 3. Discussion

Posterior cerebral artery (PCA) aneurysms are uncommon lesions, accounting for fewer than 2% of all IAs [9,10]. According to the classification system established by Zeal and Rhoton, PCA aneurysms are categorized into five groups—P1, P1/P2 junction, P2, P3, and P4 aneurysms based on their segment origin [11]. While most reported PCA aneurysms are saccular, dissecting PCA aneurysms occur more frequently in the P2 segment and at the P1/P2 and P2/P3 junctions [12,13,14]. Vertebrobasilar dissections tend to occur more often in males, whereas isolated PCA dissecting aneurysms are more prevalent in females [3,14]. Several connective tissue disorders, such as Ehlers–Danlos syndrome, polycystic kidney disease and Marfan syndrome, predispose affected individuals to arterial fragility and dissection and are therefore strongly associated with the development of dissecting aneurysms. Younger patients, particularly those aged 20 to 40 years, are more likely to present with dissecting components than purely saccular aneurysms [14,15]. PCA dissecting aneurysms may arise spontaneously or as a consequence of trauma [2,3,12]. Traumatic cerebral aneurysms account for less than the 1% of ruptured aneurysms [16,17]. Traumatic PCA aneurysms are exceedingly rare, occur predominantly in pediatric or young adult populations, and are often diagnosed in a delayed fashion due to their subtle or nonspecific initial presentation [1,2,3]. Histologically, traumatic IAs are classified as true, false, or mixed types [16,18,19]. True IAs arise from disruption of the intimal with partial involvement of deeper layers, while the adventitia remains intact, and they typically manifest as saccular lesions. False IAs—the most common—result from complete rupture of all vessel wall layers with formation of a contained hematoma. These include dissecting aneurysms, which display fusiform or pear-and-string configurations, and saccular pseudoaneurysms, which often exhibit irregular multilobulated sacs [20,21,22]. Mixed aneurysms contain both true and false components within the same arterial segment [20,23,24]. The precise prevalence of each histopathologic subtype remains uncertain, as histological confirmation is rarely obtainable in clinical case reports.

In our case, the aneurysm demonstrated a fusiform dilation with a focal, inferiorly projecting pouch arising from the left PCA P2 segment. The location and morphology of the aneurysm were particularly notable, as it extended along a relatively long arterial segment without involvement bifurcation or branch point, features that are more consistent with a false or pseudoaneurysm exhibiting fusiform characteristics. Several aspects of this presentation diverge from the typical features of spontaneous IA, suggesting an alternative pathogenic mechanism more compatible with traumatic IA formation. The report will discuss traumatic PCA dissection aneurysms, emphasizing their clinical presentation, morphology, pathological features, underlying mechanisms, associated risk factors, characteristic imaging findings, available treatment strategies, and expected prognosis.

The predominant clinical manifestation of traumatic PCA dissection aneurysms included sudden severe headache, impaired consciousness, or seizure, as well as PCA-territory ischemia. The resulting neurological deficits may involve memory dysfunction, oculomotor palsy, or visual impairments such as homonymous hemianopsia, depending on the specific PCA segments involved [2,14].

The mechanism of the traumatic PCA dissection aneurysm is generally attributed to direct mural injury caused by acceleration–deceleration forces, during which the PCA segment impacts the sharp edges of the tentorium. Such injuries typically occur in high-energy events, including falls from height, head-impact incidents, and motor vehicle collisions. The resulting tentorial-edge contact may produce focal vessel wall disruption and subsequent dissection or pseudoaneurysm formation [2,16,25].

Traumatic PCA dissection aneurysms are often not recognized early due to delayed neurologic deterioration, late rebleeding, cerebral vasospasm, and the overall rarity of the condition, which contributes to initial diagnostic oversight [1,2]. Essibayi, M. A. et al. reported that median interval from TBI to diagnosis was nine days, with a range from 2 to 90 days [2]. In the acute setting, non-contrast brain CTs are typically the first-line imaging modality following head trauma and may reveal supportive signs of tentorial-edge injury, such as tentorial-edge subdural hematomas along the tentorium or SAH within the ambient cistern or interpeduncular cistern, which suggest a traumatic etiology for PCA dissection [1,2,25]. When vascular injury is suspected, brain CTA can be rapidly performed to better delineate cerebrovascular anatomy and to identify traumatic arterial lesions. Brain CTA may demonstrate features suggestive of traumatic PCA dissection, including a pseudoaneurysm, an intimal flap, or the characteristic “string sign” reflecting tapered luminal narrowing or occlusion [4,17]. Brain MRI is also valuable for detecting traumatic PCA dissecting aneurysm formation; however, it requires more time and greater patient cooperation. T1-weighted sequences with fat saturation are particularly useful for identifying an intramural hematoma, which typically appears as a crescent-shaped hyperintense signal along the vessel wall, with highest sensitivity observed in the subacute phase. Brain MRA, preferably with three-dimensional time-of-flight (3D TOF) sequences, can further evaluate the vessel lumen morphology for luminal irregularities, tapered narrowing, or segmental dilatation [2,16]. In cases requiring definitive diagnosis or pre-treatment planning of traumatic PCA aneurysms, DSA remains the gold standard. DSA can demonstrate characteristic features of arterial dissection, including a double lumen, delayed contrast filling, a pearl-and-string configuration, or a pseudoaneurysm sac in the PCA [1,2,14]. DSA provides precise delineation of the relationship between the aneurysm, the parent artery, and the adjacent perforator, thereby facilitating optimal treatment plan.

Conservative management of traumatic PCA dissection aneurysms is generally discouraged, given the reported morbidity and mortality rate of 40–60% and the overall unfavorable prognosis associated with non-intervention [1,2]. The anatomical configuration of the P1/P2 junction warrants particular caution because of its unique hemodynamic environment. This region is exposed to turbulent flow generated by dual inflow sources, which significantly increases the risk of re-rupture. Early identification through appropriate imaging and timely therapeutic intervention—whether endovascular or surgical—is essential to preventing potentially catastrophic complications [2,13,14].

The management of traumatic PCA dissecting aneurysms is particularly challenging because of their rarity, the inherent fragility if the diseased vessel wall, and their frequent occurrence in deep-seated segments such as P2 or P3 [2,13]. Microsurgical options, including direct clipping or parent artery occlusion (PAO) with or without bypass, can provide definitive aneurysm exclusion. However, their applicability is often limited by the deep anatomical location, narrow operative corridors, and the proximity of critical neurovascular structures, including the brainstem, thalamus, and midbrain perforators [11,12,26]. PAO, performed surgically or via endovascular trapping, remains an established and effective treatment when collateral circulation through the posterior communicating artery (PComA) is adequate. Although PAO achieves high occlusion rates, it carries the risk of ischemic infarction in distal PCA territory, especially in patients with hypoplastic or absent PComA [13,14,15]. In contrast, rEVT—such as stent-assisted coiling or flow-diverting stents—offers a less invasive approach that preserves antegrade flow within the parent artery. These techniques are particularly advantageous for fusiform or dissecting aneurysms lacking a discrete neck and for patients with insufficient collateral circulation [13,15]. However, ruptured posterior cerebral aneurysms treated with rEVT have been reported to exhibit higher recanalization rates compared to microsurgical intervention or PAO [15]. The use of flow diverters in distal PCA segments remains off-label, technically challenging, and supported by limited evidence of traumatic cases [13,27,28]. Flow diverters stent for IAs’ treatments carries several well-recognized risks. Periprocedural complications encompass thromboembolic events, in-stent thrombosis, and ischemic stroke resulting from inadequate antiplatelet response or premature cessation of dual antiplatelet therapy following SAH. Hemorrhagic complications—such as delayed aneurysm rupture or intracerebral hemorrhage—may also occur, especially in large or giant aneurysms with thin aneurysmal walls. Additional challenges in acutely hemorrhagic setting include the need for antiplatelet therapy, which may increase bleeding risk. Device-related complications, such as incomplete stent apposition, migration, or vessel perforation during deployment, also remain concerns [28,29,30]. Given these complexities, individualized treatment planning—incorporating aneurysm morphology, parent vessel dominance, collateral circulation, and hemorrhagic risk—is essential to achieving optimal outcomes in patients with traumatic PCA dissection aneurysms [13,14,31].

In our case, rEVT using a flow-diverting stent combined with adjunctive coiling was selected to preserve antegrade flow and avoid ischemic complications in the left distal PCA artery that could arise from microsurgical treatment or endovascular trapping. During the acute SAH period, a relatively conservative antiplatelet regimen was adopted. The patient received a single loading dose of clopidogrel 300 mg administered 8 h before deployment of the flow diverter, followed by maintenance therapy with clopidogrel 75 mg for one year. Aspirin or intravenous glycoprotein IIb/IIIa inhibitors were deliberately withheld to minimize the risks of aneurysm rebleeding during the acute SAH.

The findings from our case suggest that when brain CT demonstrates SAH predominantly distributed along the cisterns within the PCA pathway, or when an acute tentorial subdural hematoma with recurrent bleeding occurs after trauma, a traumatic PCA injury with an associated dissecting aneurysm should be strongly suspected. In such situations, prompt vascular evaluation with brain CTA or MRA followed by DSA is essential for early and accurate diagnosis. Early intervention for traumatic PCA aneurysm can prevent further hemorrhagic or ischemic complications and significantly improve clinical outcomes. Timely rEVT using a flow-diverting stent combined with coil embolization, may serve as an effective alternative treatment strategy in appropriately selected patients, offering parent artery preservation and favorable recovery. Close clinical and radiologic follow-up after endovascular therapy is likewise critical to ensure aneurysm occlusion durability and to detect potential delayed complications.

Managing traumatic thoracolumbar unstable burst fracture in a patient receiving antiplatelet therapy after treatment for a ruptured cerebral aneurysm presents a clinical dilemma [32,33]. The decision must balance the bleeding risks associated with spinal surgery against the heightened thrombotic risks related to recent endovascular stenting or coiling, as well as the vulnerability of the brain during the first two weeks following SAH [30]. Conversely, adopting conservative management for unstable thoracolumbar burst fracture can result in prolonged immobilization, increasing the risk of pneumonia, muscle atrophy, thrombophlebitis, and delayed neurologic deterioration. Given these considerations, we decided to proceed with surgical stabilization despite the potential risk of exacerbating cerebral edema or cerebral vasospasm [34]. Our treatment strategy for L2 unstable burst fracture with spinal canal compromise was based on the following principles. First, we delayed surgery until two weeks after the SAH, allowing sufficient stabilization of the intracranial condition. Second, we maintained a single antiplatelet regimen of clopidogrel 75 mg daily to reduce intraoperative bleeding risk while still providing protection against thromboembolism or in-stent thrombosis. Third, we selected a MISS approach to minimize soft tissue damage and lessen blood loss during the procedure [35,36]. Fourth, we maintained a mean arterial pressure remained above 80 mmHg with adequate intravascular volume to support cerebral perfusion throughout the perioperative period. Fifth, we provided intensive postoperative care with close monitoring of the patient’s neurological status and hemodynamic condition.

## 4. Conclusions

Traumatic PCA dissecting aneurysms are rare but carry substantial risks of rupture and morbidity. Vigilance for vascular injury after head trauma is essential, and careful evaluation of cerebrovascular images is crucial for early diagnosis. Based on the experience of this case, prompt rEVT with a flow-diverting stent, in combination with coil embolization, would probably be an effective alternative treatment strategy for traumatic PCA aneurysm to achieve a better neurologic outcome than PAO. Carefully timed MISS with tailored antiplatelet management may provide an effective option for thoracolumbar unstable burst fractures after acute ruptured cerebral aneurysm treatment, although substantial perioperative risks remain.

## Figures and Tables

**Figure 1 life-16-00034-f001:**
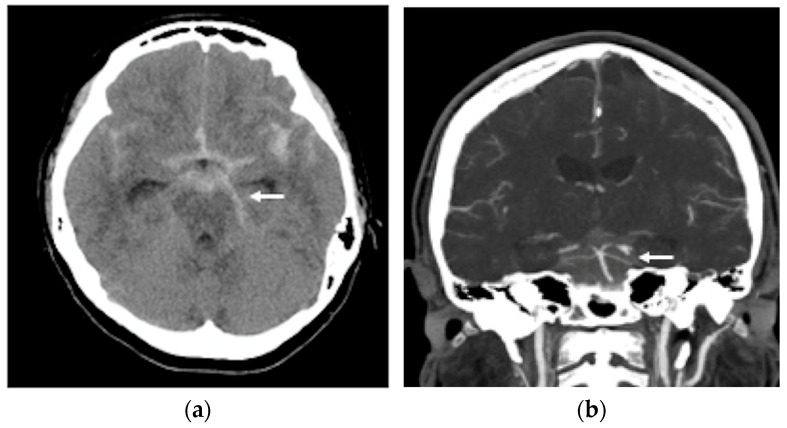
(**a**) A non-contrast brain computer tomography (CT) revealed diffuse subarachnoid hemorrhage (SAH), with a thickness exceeding 1 mm within the basal cistern, predominantly along the left side ambient cistern (white arrow head); (**b**) brain CT angiography demonstrated an irregular vessel contour in the left PCA P2 segment, with a focal arterial pouch consistent with a dissecting aneurysm with a saccular rupture (white arrow head).

**Figure 2 life-16-00034-f002:**
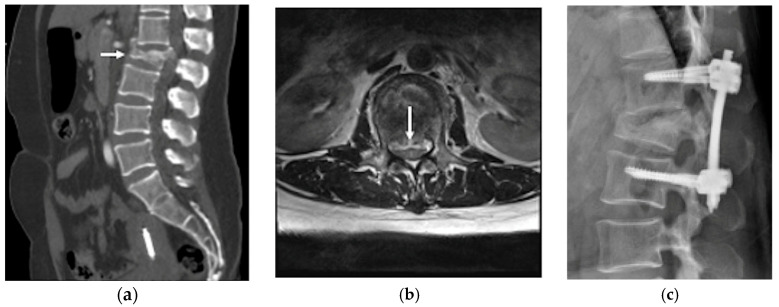
(**a**) Lumbar spine CT scan identified burst fractures in the L2 vertebral with spinal canal compromise (white arrow); (**b**) lumbar spinal magnetic resonance imaging (MRI) T2 weighted image axial view confirmed an L2 level severe spinal canal compromise, resulting in approximately 65% dural sac stenosis (white arrow); (**c**) lumbar spine lateral view showed percutaneous transpedicle screw fixation from L1 to L3.

**Figure 3 life-16-00034-f003:**
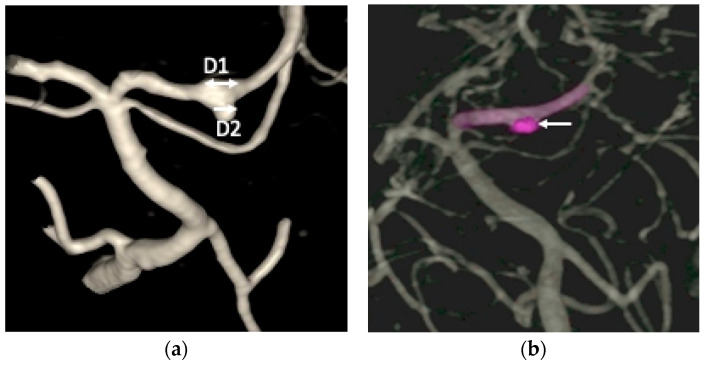
(**a**) The digital subtraction catheter angiography (DSA) identified an approximately 5 mm long fusiform dilation (D1, double arrow head) along the anterior portion of the left posterior cerebral artery (PCA) P2 segment, accompanied by an inferiorly projecting saccular pouch measuring 2.5 mm in diameter (D2, single arrow head); (**b**) the endovascular treatment resulted in near-complete elimination of contrast opacification within the previously visualized dissecting pouch, indicating successful occlusion of the false lumen of the PCA dissection aneurysm (white arrow head).

**Figure 4 life-16-00034-f004:**
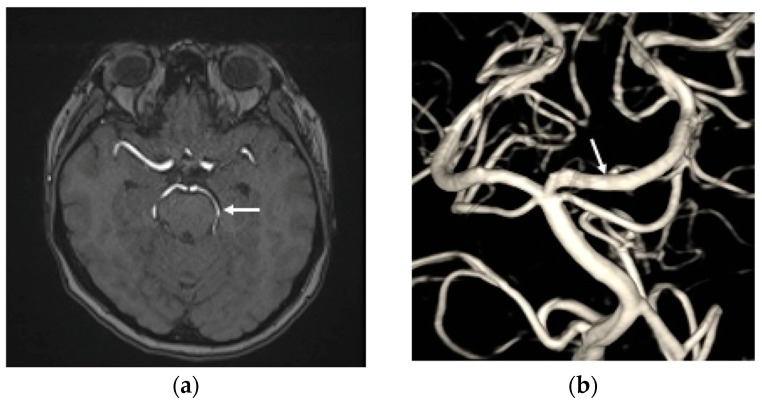
(**a**) The time-of-flight magnetic resonance angiography revealed the left PCA patency (white arrow head) without thromboembolic events post transarterial endovascular treatment (TAE) three months; (**b**) the DSA at six months after TAE shows that the left PCA P2 dissection aneurysm has achieved total obliteration, with no stenosis observed in the parent artery (white arrow head).

## Data Availability

The original contributions presented in this study are included in the article. Further inquiries can be directed to the corresponding author.

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
