# Peer review of "Ruptured Posterior Cerebral Artery Dissecting Aneurysm After Trauma: A Case Report and Literature Review"

_life, 2025, doi:10.3390/life16010034_

Round 1
Reviewer 1 Report
Comments and Suggestions for Authors
This manuscript presents a well-documented case of a ruptured traumatic posterior cerebral artery (PCA) dissecting aneurysm treated with flow diversion and stent-assisted coiling. The case is clinically relevant due to the rarity of traumatic PCA aneurysms and the increasing application of flow-diverting devices in distal circulation. The report is generally well structured, but certain sections require clarification, additional referencing, and rewriting for grammatical precision and improved fluency. The discussion is thorough, though it would benefit from more explicit emphasis on decision-making rationale, especially regarding antiplatelet management in the setting of acute subarachnoid hemorrhage. The conclusions are appropriate and consistent with the case details.
Specific Remarks:
-The sequence from initial presentation to endovascular treatment should be described more explicitly. Specifically, the reason for delaying endovascular intervention until day 3 after SAH must be clarified, as timing is clinically significant in traumatic aneurysm management.
-The decision to administer a 300 mg clopidogrel loading dose prior to flow diverter placement after recent SAH should be better justified, as antiplatelet therapy in acute hemorrhagic settings entails specific risks. Please explain perioperative monitoring and hemorrhagic risk considerations.
-The discussion should more directly compare the flow-diverting approach with parent artery occlusion in this anatomical context, including why artery occlusion was not favored in this case (caliber? collateral vessels? other risk?).
Author Response
Dear Reviewer:
Thank you very much for your suggestions and comments on our manuscript. Please see the attachment.
Best Regards
Dr. Jack Yang
Department of Neurosurgery, CSMH, Taiwan, China

Reviewer 2 Report
Comments and Suggestions for Authors
You present multiple images but fail to analyze them properly. You provide exactly TWO measurements in the entire report (5mm length, 2.5mm diameter) and nothing else. Where are the quantifications of SAH volume, spinal canal stenosis percentage, vessel caliber changes, or degree of obliteration? You merely describe what you see without any numerical rigor.
Your figures lack annotations, arrows, scale bars, or marked regions of interest. Figure legends are simple descriptions, not analytical explanations. This reads like a radiology report, not a scientific case study.
Your introduction is bloated and unfocused. You fail to clearly state upfront what makes this case report-worthy or novel. Get to the point: what is unique about YOUR patient?
Your case presentation is disorganized. Clinical details are scattered randomly. You mention "consciousness level" but provide no Glasgow Coma Scale score. You state "vital signs were stable" without specifying what they were.
Timing is vague, "after SAH for 3 days" is unclear phrasing. Where is the systematic neurological examination? You cannot expect readers to identify similar presentations when you provide such incomplete clinical documentation.
Your data reporting is purely descriptive with zero analysis. You perform multiple imaging studies but never explain HOW you measured anything, assessed image quality, or validated findings.
No discussion of inter-observer agreement, measurement protocols, or technical parameters.
You mention modified Rankin Scale once at the end; why not document it systematically throughout?
Your conclusions wildly overreach. You claim "effective strategy" and "markedly enhanced outcomes" based on ONE patient. Yes, this is a case study, but still make sure that when using terms such as "markedly enhanced" you also specify "for this patient" in the same sentence. And add a point in the conclusion to reiterate that this a case study.
Author Response
Dear Reviewer:
Thank you very much for your suggestion and comment. Please see the attachment of author's reply.
Best Regards
Dr Jack Yang
Department of Neurosurgery, CSMH, Taiwan, China

Reviewer 3 Report
Comments and Suggestions for Authors
The authors report a case of posterior aneurysm that ruptured and derived after an accident.
The introduction is ok but should include the prevalence of dissecting aneurysms in cases of trauma and non trauma. Thats the whole point of the work and should be more thoroughly discussed.
The quality of the text should be improved throughout the text.
From your text i understand that the patient with traumatic SAH and ruptured aneurysm was loaded with blood thinner and was observed for three days. You mean after endovascular treatment, right? Because otherwise it doesnt seem quite right. I understand a pre-endovascular loading. However, recent literature regarding ruptured brain aneurysms talks about a fast treatment with 1 day if possible or within 3 in case it is not otherwise possible because of the high re-bleeding rates but still not with clopidogrel treatment on top. That was very risky.
Also very risky was to do a spine surgery in a patient with SAH. The prone position and the art of surgery are not very brain friendly.
About the spine surgery, I would not stop a stabilization at the thoracolumbal junction due to high risk of adjacent segment instability in the near future (Th12-L1). Furthermore, I would expect the rods to have a different curvature, the screws to be de a little deeper in the vertrabra (these two reasons will soon lead to loosening of the upper screws) and most importantly to make a decompression of the nerves as they are obviously compressed. You describe a hemilaminectomy but we dont see it in the pictures so please provide other pictures showing an appropriate decompression of the spinal canal.
You should refer to increased vessel dissection and aneurysm rates in patients with genetic and connective tissue disorders (e.g. Ehler Danlos, Marfan syndrome etc). Can we be sure that your patient didnt have such?
In the discussion I cannot understand why your aneurysm case is a true dissective one and not a pseudoaneurysm after traumatic vessel dissection.
In the beginning of the discussion part you repeat yourself a lot with the rest of the text so please remove some things to make the text shorter and more concentrated/consistent.
In addition, the general knowledge in discussion should be omitted and you should concentrate mostly on cases like yours. Tell us the right diagnostics and not the possible ones, the proper reaction and treatment, point out the highlights and pitfalls in such cases and give us a message, the purpose of your paper. I find your management paragraph for example very good and you should keep it. But most of the info before it is unnecessary. Just tell us to do CT in emergencies, where we are going to see a bleeding in such cases, do a CT angiography or a DSA and thats all. You write it in your last paragraph of the discussion, why lose so many words in lines 179-198.
The symptoms of the ruptured aneurysm are the same, either traumatic or not, with or without dissection.
You should also write a few words about expertise needed in such cases (neurovascular centers) and the importance of interdisciplinary cooperation.
Finally, similar problem with dissected vessels we have oftenly in vertebral arteries and cervical trauma. These should be examined too after a severe accident.
Comments on the Quality of English Language
Pretty low quality, should be profoundly improved.
Author Response
Dear Reviewer:
Please see the attachment. Thank you very much for your review and comments.
Best Regards
Dr Jack Yang
Department of Neurosurgery, CSMH, Taiwan, China

Round 2
Reviewer 2 Report
Comments and Suggestions for Authors
In my first review, I noted: "Your data reporting is purely descriptive with zero analysis... You perform multiple imaging studies but never explain HOW you measured anything... No discussion of inter-observer agreement, measurement protocols, or technical parameters." You responded by stating: "Because we only have a single case, we cannot conduct intergroup analysis." This response indicates a fundamental misunderstanding of the critique. I was not requesting a statistical "intergroup analysis" or a comparison of patient populations, which is indeed impossible in a single case report. Instead, I was requesting methodological transparency regarding the specific quantitative data presented for this individual patient. When reporting precise figures such as a "5 mm long" aneurysm and "2.5 mm diameter," scientific rigor requires defining the technical parameters (e.g., the specific 3D-DSA software tools used, the plane of measurement) and confirming the validity of these observations (e.g., stating that a second neuroradiologist independently reviewed and confirmed these specific dimensions). Please revise the manuscript to include a brief description of how these measurements were technically derived and validated, distinguishing this methodological rigor from statistical power.
Reviewer 3 Report
Comments and Suggestions for Authors
Traumatic cerebral aneurysms account for less than the 1% of ruptured aneurysms. (Traumatic intracranial aneurysm in blunt trauma Bardiya Zangbar, Julie Wynne, Bellal Joseph, Viraj Pandit, David Meyer, Narong Kulvatunyou, Mazhar Khalil, Terence O’Keeffe, Andrew Tang, Michael Lemole, Randall S. Friese, & Peter Rhee)
I understand your arguments with the blood thinner and the spine procedure and it was indeed a very difficult decision, but it is not right to do a spine surgery in a patient with a ruptured aneurysm. So you should make your self critic in the manuscript because you endangered patient's life. My intention is not to criticize you but to make clear in the text that is not right. The patient could have died or get irreversible brain damage during the spine surgery. He had brain oedema and SAH from the ruptured aneurysm and the literature says to keep minimum manipulation of those patients.
Let a native speaker to re-read and correct your text.
In your case description you say that Clopidogrel loading succeeded 8 hours before treatment, in the discussion 24 hours before treatment.
Comments on the Quality of English LanguagePretty low quality, should be profoundly improved.
Round 3
Reviewer 3 Report
Comments and Suggestions for Authors
With the changes done it improved a lot.
Author Response
Dear Reviewer:
Thank you for your advice and suggestions for the manuscript. We made final revision of the manuscript and improved the quality of figures and English.
Best regards,
Jack Yang